# Strain Typing of Classical Scrapie and Bovine Spongiform Encephalopathy (BSE) by Using Ovine PrP (ARQ/ARQ) Overexpressing Transgenic Mice

**DOI:** 10.3390/ijms23126744

**Published:** 2022-06-16

**Authors:** Olanrewaju I. Fatola, Markus Keller, Anne Balkema-Buschmann, James Olopade, Martin H. Groschup, Christine Fast

**Affiliations:** 1Neurosience Unit, Department of Veterinary Anatomy, University of Ibadan, Ibadan 200005, Nigeria; fatolan@yahoo.com (O.I.F.); jkayodeolopade@yahoo.com (J.O.); 2Friedrich-Loeffler-Institut, Institute of Novel and Emerging Infectious Diseases, 17493 Isle of Riems, Germany; markus.keller@fli.de (M.K.); anne.balkema-buschmann@fli.de (A.B.-B.); martin.groschup@fli.de (M.H.G.)

**Keywords:** classical scrapie, atypical scrapie, BSE, ovine BSE, strain typing, transgenic mice, PrP^Sc^, immunohistochemistry

## Abstract

Transmissible spongiform encephalopathies (TSE), caused by abnormal prion protein (PrP^Sc^), affect many species. The most classical scrapie isolates harbor mixtures of strains in different proportions. While the characterization of isolates has evolved from using wild-type mice to transgenic mice, no standardization is established yet. Here, we investigated the incubation period, lesion profile and PrP^Sc^ profile induced by well-defined sheep scrapie isolates, bovine spongiform encephalopathy (BSE) and ovine BSE after intracerebral inoculation into two lines of ovine PrP (both ARQ/ARQ) overexpressing transgenic mice (Tgshp IX and Tgshp XI). All isolates were transmitted to both mouse models with an attack rate of almost 100%, but genotype-dependent differences became obvious between the ARQ and VRQ isolates. Surprisingly, BSE induced a much longer incubation period in Tgshp XI compared to Tgshp IX. In contrast to the histopathological lesion profiles, the immunohistochemical PrP^Sc^ profiles revealed discriminating patterns in certain brain regions in both models with clear differentiation of both BSE isolates from scrapie. These data provide the basis for the use of Tgshp IX and XI mice in the characterization of TSE isolates. Furthermore, the results enable a deeper appreciation of TSE strain diversity using ovine PrP overexpressing transgenic mice as a biological prion strain typing approach.

## 1. Introduction

Transmissible spongiform encephalopathies (TSEs), a group of progressive and fatal neurodegenerative diseases, were described in a number of species, for example, scrapie in sheep and goats and bovine spongiform encephalopathy (BSE) in cattle. The latter is a zoonotic disease, causing the variant Creutzfeldt Jacob disease in humans [1,2,3,4,5]. The TSE causative agent was postulated to be solely proteinaceous in nature [6] and to induce the conversion of the normal host-encoded prion protein (PrP^C^) into the pathogenic and partially protease-resistant forms (PrP^Sc^).

Two types of scrapie have been reported so far: classical and atypical scrapie [7]. While classical scrapie is infectious, atypical scrapie is regarded as rather sporadic and non-contagious, rather than a contagious disease [8]. Furthermore, natural BSE infections have been diagnosed in two goats so far [9,10,11]. In contrast, in sheep, only experimental BSE infections are known [12,13,14,15,16].

Classical scrapie is reportedly diverse in its traits [17], and strain typing has become very essential in determining the nature of the infectious agent [18]. A scrapie strain is defined by its phenotypic characteristics in the natural host, including clinical signs, lesion and PrP^Sc^ profile and the biochemical PrP^Sc^ characteristics (e.g., protease sensitivity, molecular migration in Western blot, glycosylation pattern). Moreover, the biological characteristics (incubation period, attack rate, lesion and PrP^Sc^ profiles, biochemical PrP^Sc^ pattern) induced in experimental hosts, in particular rodent models, have become increasingly important for strain discrimination [19].

Previously, the inoculation of diverse scrapie isolates into a variety of conventional wild-type mice, including the RIII line, has enabled the discovery of more than a dozen strains of scrapie agents [20,21,22]. In addition, linking the causative agent of cattle BSE to that of the human vCJD has evidently been proven using these experimental mouse models [4,5]. However, the species barrier may hamper the use of conventional mouse models, resulting in prolonged incubation times or reduced transmission rates, in some cases even in the inability to observe end-stage disease with sheep scrapie isolates in the first passage. In recent years, transgenic mouse models overexpressing the homologous host prion protein have been used effectively to address this particular obstacle [23,24,25]. Therefore, the full characterization of isolates has evolved from the traditional use of a combination of different wild-type mouse lines to the use of transgenic mouse lines overexpressing the prion protein from different species and genotypes. However, it has to be borne in mind that there is no basis yet to directly compare TSE strains identified in conventional and transgenic mouse models [19].

Yet to be standardized is the strain typing procedure using transgenic mouse models. In this regard, the immunohistochemical PrP^Sc^ profile and the incubation period in combination with Western blotting (WB) were proposed as means of standardization [26,27,28]. Recently, Nonno et al. (2020) [29] introduced the relative transmission efficiency of TSE isolates as a biological signature of the isolates across several mouse models. This approach in combination with PrP^Sc^ typing in the recipient hosts allows the examination of the biological properties of a certain TSE isolate by their specific interaction with different mouse models.

Previous strain typing studies in transgenic mice have identified at least five prion strains responsible for classical scrapie [19,30,31], but the exact number is unknown and most field isolates even contain sub-strains at different proportions [29,32]. Evidently, some isolates are not completely stable and different replication environments might play an important role in defining their biological properties, which can shift on transmission and even affect the ability of certain strains to cross the species barrier [19], as was shown for some classical scrapie isolates in humanized transgenic mice [33]. Such divergence may explain the high diversity of classical scrapie strains.

Therefore, on this background, the use of transgenic mice overexpressing ovine PrP^C^ is regarded as one of the alternative approaches for strain typing [7]. Transgenic mouse lines (Tgshp IX and Tgshp XI) overexpressing both the wild-type ARQ allele of the ovine PrP are regularly used [29,34], but standardization and comparison of these mouse models are yet missing. Therefore, well-defined scrapie strains originating from different sheep genotypes as well as cattle and sheep BSE were passaged in both transgenic mouse models aiming toward standardizing this method.

## 2. Results

In this study, transmission characteristics of different, well-known classical and atypical scrapie, as well as BSE and ovine BSE isolates, were evaluated in two transgenic mouse models (TgshpIX and TgshpXI, both ARQ/ARQ) to explore the input of transgenic mice in prion strain typing.

### 2.1. Incubation Period

All isolates were transmitted to Tgshp IX and XI mice with an attack rate of almost 100% (in total 97%). However, differences in incubation time were obvious between the isolates and between the mouse models (Table 1).

All ARQ isolates produced the lowest incubation periods, which were very similar in both mouse models, with S805 (160 ± 18 dpi in Tgshp IX and 158 ± 20 dpi in Tgshp XI), LAN (190 ± 3 dpi in Tgshp IX and 195 ± 11 dpi in Tgshp XI) followed by the ARQ ovBSE isolate (230 ± 63 dpi in Tgshp IX and 231 ± 9 dpi in Tgshp XI). As expected, the VRQ isolate DAW induced a much longer incubation period (374 ± 44 dpi in Tgshp IX and 342 ± 99 dpi in Tgshp XI) as compared to the ARQ scrapie isolates, even showing a slight difference between the mouse models used. The original atypical scrapie isolate was a 1% homogenate and therefore higher diluted as compared to the other isolates. Nevertheless, a relatively short incubation time of 272 ± 32 dpi with atypical scrapie was observed in Tgshp XI mice, which is in clear contrast to the comparably long incubation period in Tgshp IX mice (474 ± 7). The cattle BSE isolate induced an incubation period of 351 ± 165 dpi in Tgshp IX mice. However, high variability was seen in this group, and only one mouse showed a very high incubation period of 677 days leading to the high standard deviation of the mean, without this animal the mean incubation period would be much lower (286 ± 86). Moreover, an additional BSE isolate (BSE 2) processed in Tgshp IX mice induced a similar low incubation period of 262 ± 21 dpi. Most interestingly, in contrast, the incubation periods seen in Tgshp XI mice were almost doubled (445 ± 185 dpi) as compared to Tgshp IX and close to the results obtained in RIII mice (data not shown).

### 2.2. Lesion Profile

Traditionally, nine grey matter and three white matter neuroanatomic regions are examined by histopathology for lesion profiling in experimental mouse transmission studies [35].

The overall frequency of spongiform lesions in TgshpIX, and TgshpXI mouse models were close (33% and 36%, respectively), but a higher degree of variation in the level of the spongiform lesion was seen in TgshpXI (10.0%) as compared to TgshpIX (5.7%) mice.

However, considering the quality of lesions, both mouse models did not provide a clear difference for the examined isolates. The only distinct discriminating pattern in both mouse models was the differences in corpus callosum (CC) plaque formations, which were only induced by ARQ and BSE isolates. The ARQ isolates (LAN, S805 and ovBSE) and BSE2, unlike the VRQ isolate (DAW) and AS, induced the formation of plaques in the corpus callosum (Appendix A).

### 2.3. PrP^Sc^-Profile

In Table 1 the most important results of the immunohistochemical PrP^Sc^ deposition pattern in the mouse brain that allowed the discrimination of the different isolates are summarized.

The PrP^Sc^ profiles of BSE and ovBSE in Tgshp IX mice are very similar with main peaks at regions 1, 4, 5, (6) and 2* and CC. A slight difference is seen only in region 6, which is less involved in ovBSE as compared to BSE (Figure 1). Distinct PrP^Sc^ profiles are seen with all scrapie isolates discriminating the isolates from BSE/ovBSE as well as from each other. A unique PrP^Sc^ profile, however, was seen with atypical scrapie, in which all mice were distinctly positive in the molecular layer of the cerebellum.

The PrP^Sc^ profile of BSE and ovBSE in Tgshp XI mice is very close and grey matter peaks 1, 4 and 5 are similar to the results observed in Tgshp IX mice. However, white matter peaks are seen in regions 1*, 2* and 3* (Figure 2). Distinct PrP^Sc^ profiles were seen with all scrapie isolates discriminating the isolates from BSE/ovBSE as well as from each other. Moreover, besides few congruencies, the results induced by the classical scrapie isolates mostly differ from the profiles observed in Tgshp IX mice, but not for atypical scrapie. The latter induced the same unique staining reaction as described for Tgshp IX mice with a distinct PrP^Sc^ accumulation in the molecular layer of the cerebellum. However, individual mice also showed weak accumulation in regions 2, 6 and in CC with atypical scrapie isolates (Figure 2).

Besides the standard procedure described by Fraser and Dickinson (1968) [35] for strain typing in conventional mice, it is worth targeting additional regions in the Tgshp IX and XI mouse models for discriminating different isolates.

Cerebellum (Figure 3A–C): the cerebellum is a very helpful tool in both mouse models, not only for characterizing atypical scrapie, which is the only isolate inducing PrP^Sc^ accumulation in the molecular layer (Figure 3A), but also to clearly discriminate BSE/ovBSE, which always induce PrP^Sc^ accumulation in the granular layer and white matter in both mouse models (Figure 3B). Whereas this pattern is unique in Tgshp IX mice, in Tgshp XI mice this PrP^Sc^ deposition pattern is also induced by DAW. On the other hand, the two classical scrapie ARQ isolates, S805 and LAN, induce very little (Tgshp IX, Figure 3C) or no PrP^Sc^ (Tgshp XI) in cerebellar white matter, and the molecular and granular layer remain completely free of deposits.

Corpus callosum (CC, Figure 3D–F): most isolates induced a marked PrP^Sc^ accumulation (Figure 3D,E) in both mouse models, in a coalescing or even plaque and plaque-like pattern. However, atypical scrapie and DAW (Figure 3F) induced only a small amount of PrP^Sc^, if any, mainly in Tgshp XI.

Hippocampus (Figure 3D–F): most isolates induced a variable multifocal staining pattern, but not for atypical scrapie and S805, which are devoid of any PrP^Sc^ accumulation in Tgshp IX mice. In contrast, in Tgshp XI, a weak amount of PrP^Sc^ was induced by S805. Even more promising are the fimbriae hippocampi which are distinctly affected by BSE/ovBSE in both mouse models (Figure 3D, arrowhead), only LAN also showed in single Tgshp IX mice as a weak PrP^Sc^ deposition.

Thalamus: discriminating patterns were visible, but with distinct differences between the mouse models. For example, in Tgshp IX mice, all ARQ isolates induced a multifocal reaction pattern, whereas with DAW, a diffuse PrP^Sc^ distribution involving all nuclei was prominent. On the other hand, in Tgshp XI mice a multifocal pattern was seen for all isolates but for S805, the latter only affected the dorsolateral nuclei.

A region worth mentioning is the corpus striatum (CS), where a distinct PrP^Sc^ accumulation was only seen with LAN in both mouse models, whereas for BSE/ovBSE, DAW and S805 only, single positive cells were observable if any at all.

On a cellular level, the PrP^Sc^ reaction pattern showed some remarkable differences, which are helpful tools in discriminating isolates by means of immunohistochemistry in both models. A subpial reaction (SBPL) pattern, for example, was widespread, profound and significant (Appendix A) in BSE in both models, involving almost all examined regions, in particular the brain stem, cerebellum, midbrain, diencephalon (i.e., *fimbriae hippocampi*) and most parts of the cerebrum (Figure 4A1). In both mouse models, such a pattern was also seen with LAN, but to a much lesser degree and extension (Figure 4B1). In all other isolates, this reaction pattern was only rarely seen in a few regions and not in all mice (Figure 4C1).

Another distinct reaction pattern is the detection of plaques and plaque-like formations in both models, which were prominent, widespread and significant in BSE, involving all regions examined (Appendix A) in particular, the midbrain (Tgshp IX, Figure 4A1,A2) and CC (Tgshp XI). This PrP^Sc^ pattern is also prevalent in both models infected with LAN, but to a much lesser degree and not in all mice and not in all regions (Appendix A). LAN induced the most variable reaction pattern instead, with a coarse granular to coalescing deposition as the most prominent, mixed up with some specific patterns, for example, a perivascular accumulation (Figure 4B1,B2).

An additional unique pattern was found in both mouse models with S805 in which a very prominent intraneuronal/intraglial PrP^Sc^ (ITNR/ITGL) accumulation was visible, associated with a very mild fine granular staining reaction (Figure 4C1,C2). In contrast, all other strains revealed an equal mixture of intracellular and variable extracellular PrP^Sc^ accumulations with fine up to coalescing depositions.

## 3. Discussion

Prion strain properties depend on the specific conformation of PrP^Sc^ [32] and, among others, are defined by their disease properties induced in rodent models, including attack rate and incubation period as well as lesion and PrP^Sc^ profiles in the brain. Besides conventional mouse models, nowadays, the characterization of isolates includes the use of transgenic mouse models devoid of murine but instead overexpressing the prion protein of different host species. However, the analysis of transgenic mice in TSE strain typing has not been standardized yet. Data generated from previous work which used different transgenic mouse lines showed that ovinized transgenic mice are quite promising, not just for characterizing scrapie but also for BSE isolates [26,36,37,38].

In the study presented here, two transgenic mouse lines, used in different laboratories in Europe, that overexpress ovine PrP (ARQ/ARQ) 2–4-fold (TgshpIX) and 4–8-fold (TgshpXI), respectively, were inoculated intracerebrally with BSE, ovineBSE and well-defined isolates of sheep scrapie originating from different host genotypes. To achieve standardization of prion strain typing in these mouse models, we analyzed the induced disease phenotypes in the mice. The data generated showed that the two transgenic mouse models were fully susceptible to both cattle and sheep PrP^Sc^ isolates, which provoked nearly 100% attack rates. We further noted variations in the incubation periods displayed by the isolates that were genotype related. The brain lesion profiles induced by the isolates in both models generated no compelling differences when compared to each other. However, the PrP^Sc^-profiles in the brains of both transgenic mouse lines allowed for clear discrimination of the different TSE isolates investigated.

### 3.1. Incubation Period

The conversion of PrP^C^ to PrP^Sc^ and the subsequent accumulation of PrP^Sc^ is the key event for the successful transmission of a TSE isolate. This process is mainly influenced by the amino acid sequence and therefore by the conformation of the prion protein from both the host (PrP^C^) and donor (PrP^Sc^) [39]. This also includes the ability of certain prion strains to cross the species barrier [40]. Thus, the more similar PrP^C^ and PrP^Sc^ are, the easier the transmission will be. Once transmitted, the disease progresses and the interval between the initial infection and the death of the animal defines the incubation period [41]. The transgenic mouse models used here displayed differences in the incubation period, which are clearly due to the species barrier phenomenon expressed by the different amounts of PrP^C^ in the two models, but also in the genotype of the TSE inocula and the mouse model. In this regard, it comes as no surprise that we observed very low incubation times in both mouse models inoculated with ARQ classical scrapie isolates (LAN, S805) when compared to the VRQ (DAW) and ARR (atypical scrapie) isolates. This evidence could be attributed to the fact that both transgenic mouse lines also expressed the ARQ allele of the ovine PrP, further elaborating the combined influence of PrP genotypes of both the host and donor on TSE phenotypes. Thackray and his team have, in a similar manner, observed enhancement of TSE transmission in ovine PrP transgenic mice inoculated with an isolate of the same genotype [38].

Interestingly, the elevated incubation period documented for the DAW (VRQ) and atypical (ARR) scrapie isolates also presented notable differences between the transgenic mice. In the TgshpIX mouse model, the DAW and atypical scrapie inocula elicited mildly longer periods relative to the TgshpXI mouse line. This could be a result of the higher expression of the scrapie prion protein gene in TgshpXI (4–8 fold) mice relative to TgshpIX mice (2–4 fold), facilitating the progressing conversion once the transmission was achieved.

However, to our surprise, the opposite was the case with BSE where a distinctly longer incubation period was noted in the TgshpXI mouse line (445 ± 185 dpi) compared to the TgshpIX mouse line (BSE: 351 ± 165 dpi, BSE2: 262 ± 21 dpi). In this regard, it is worth mentioning that the difference seen between the incubation period of BSE and BSE2 in Tgshp IX depends on one individual, a calculation without that animal would result in quite similar incubation periods. These results indicate that the TgshpXI mice were less susceptible to BSE than the TgshpIX mice. Additionally, with regards to all other isolates in the transgenic mice, the BSE produced the longest incubation period in TgshpXI mice. It is probable that pathological conversion of PrP^C^ by BSE in particular might have been hampered in TgshpXI mice as a result of the relatively higher ovine PrP^C^ overexpression. This could mean that BSE encountered a more potent species barrier in TgshpXI than in TgshpIX mice. Furthermore, this unusual course of BSE in the TgshpXI mouse line perhaps indicates that there could be undefined host-derived cellular factors, i.e., neurotransmitters and/or different cytokines that influence prion susceptibility.

In contrast, the incubation period of the ovine BSE isolate produced not only much shorter incubation periods in both transgenic mice when compared to BSE, but the incubation periods were also very similar in both mouse models. Besides the obvious congruencies of genotype and host species with the mouse models used here, which clearly facilitate transmission and propagation, it has to be borne in mind that a transmission study using humanized transgenic mice has linked a more aggressive nature of both ovine and caprine BSE, in contrast to cattle BSE, to a significant modulation by the ovine prion protein primary sequence [42].

The primary structure of the host PrP^C^ is a crucial factor for successful prion transmission [43] and the results induced by the sheep isolates are easily explained by a match or mismatch between the amino acid sequences of the prion protein in the host and donor species that determine susceptibility or resistance to TSE [40]. Therefore, the protracted transmission of DAW (VRQ) and AS isolates in both transgenic mouse lines could be a result of poor matching of the mice’s PrP amino acid sequences with the ovine PrP amino acid sequences. The reverse could logically hold true in the transgenic mice following inoculation with ARQ scrapie or ovine BSE isolates. In these mouse lines, overexpression of the ovine cellular prion proteins facilitates their transformation as a result of suitable matching of their amino acid sequences. Of particular interest is the distinctly stronger species barrier of Tgshp XI mice for the classical BSE isolate, indicating that this mouse line is less suitable for transmission experiments of isolates from other species than sheep/goats.

### 3.2. Lesion Profile

TSE strains are traditionally differentiated by the lesion profile scoring of 12 defined brain regions [21,35,44]. Lesion profiling is basically a semiquantitative method that provides a basis for discriminating between the various prion agent strains.

Unfortunately, the lesion profiles observed here did not provide robust differences for all isolates. This observation appears to be corroborated by earlier research works, which do not favor the use of histopathology to characterize TSE in the natural host [45,46,47,48].

The work carried out by Crozet and his team in an ovinized transgenic mouse line put emphasis on discriminating ovine BSE from classical BSE using the ovine BSE-associated florid plaques [36]. Likewise, in spite of no compelling discriminating pattern of HE lesions, plaque formation in our transgenic mice also provided genotype-based discrimination of scrapie isolates examined here. Only the VRQ-isolate DAW and the ARR-isolate atypical scrapie failed to produce plaques at the CC in both models. Previous work has also established no plaque formation following exposure of ARQ ovine transgenic mouse lines to ARQ/VRQ scrapie isolates [49]. Therefore, plaque formation observed only with the ARQ isolates in the transgenic mice could further mean that identical PrP genotype between the donor and recipient host species will more likely facilitate the generation of plaques. However, other factors must also be involved as plaque formation is a distinct pattern of BSE, which is an isolate from a different species.

### 3.3. PrP^Sc^ Profile

Immunodetection of PrP^Sc^ is regarded as a more effective means of identifying TSE strains even when neuropathology is not visible [50]. In the past, several approaches have been carried out focusing mainly on the standardized method established by Fraser and Dickinson [35].

The immunohistological data revealed differences in the ability of each transgenic mouse model to discriminate the TSE isolates. Despite the common genetic background shared by both transgenic mouse models, the partially different neuroanatomical patterns of PrP^Sc^ accumulations detected with the same isolate demonstrate the importance of ovine PrP^C^ expressed at different levels in these two models. Therefore, how easily PrP^C^ will be converted to PrP^Sc^ in a specific neuroanatomical area might depend on the quantity of PrP^C^ at that site. Moreover, the PrP^Sc^ overall deposition occurs at higher amounts in TgshpXI than in the TgshpIX mouse model even though both models displayed practically the same attack rates. An obvious reason is the higher PrP^C^ expression levels in TgshpXI (4–8 folds) relative to TgshpIX (2–4 folds), which might promote easier and more propagation of PrP^Sc^ but may also potentially blur the differences between the different brain sites examined.

In both transgenic models, we observed unique PrP^Sc^-profiles, discriminating each isolate from the rest, but most importantly identifying the BSE isolates. As was shown for sheep brains, accumulation of PrP^Sc^ occurs at different morphological, topographical and cell-associated patterns [51]. Thus, we learned that taking additional regions into account to a great extent, facilitated clear discrimination of the various TSE isolates investigated. In a previous study, such an approach clarified certain scrapie isolates with an almost indistinguishable lesion profile from BSE as distinctly different from BSE [52]. Hence, in addition to the brain regions traditionally evaluated for TSE strain typing [35], certain sites were also analyzed for characteristic PrP^Sc^ aggregation patterns. These regions include the different compartments of the cerebellum (molecular layer, granular layer) which show unique patterns not only for atypical scrapie but also for BSE and ovBSE. An additional typical accumulation site for BSE/ovBSE is the *fimbria hippocampi*. On the other hand, CC is involved only in animals with an ARQ background, CS showed a distinct reaction pattern only for LAN and S805 is the only classical scrapie isolate which does not induce a PrP^Sc^ deposition in the hippocampus. Taken together, every strain has a characteristic pattern and we were able to discriminate between BSE and scrapie isolates as such but also defined classical scrapie strains from each other by using this strategy.

A very characteristic pattern is particularly seen with atypical scrapie, which failed to generate PrP^Sc^ aggregations in the majority of the standardized brain regions of both transgenic mouse models, but uniquely induced PrP^Sc^ accumulations in the cerebellar molecular layer of both mouse models. Moreover, atypical scrapie showed distinct stability as our results are in concordance with a recent report of PrP^Sc^ deposits within the cerebellum, hippocampus and corpus callosum of TgshpXI mice inoculated with AS-infected brain samples from Portugal [34]. However, hippocampal deposits noted in this study were faint if nothing at all, and was shown in Tgshp XI only, supporting our theory that a higher PrP expression pattern facilitated the conversion.

Besides the site of accumulation, the cellular PrP^Sc^ reaction pattern offers an additional tool to discriminate the isolates. For instance, plaque formation in the brains of transgenic mouse models overexpressing the ovine PrP reportedly served as a basis for differentiating ovine-adapted BSE from ovine scrapie [36]. In this present study, significant plaques (PL) and plaque-like depositions were associated with BSE, a distinct and widespread distribution was also seen with ovBSE. In contrast, in ARQ isolates, this pattern was confined to the CC and with atypical scrapie to the cerebellum. Another reaction pattern showing discriminatory abilities is the significant subpial (SBPL) reaction pattern induced by BSE, which is also distinctly seen in mice inoculated with ovBSE, mildly induced by LAN and only rarely seen with all other isolates. Additionally, a unique staining reaction, which is confined to an intraglial (ITGL) and intraneuronal (ITNR) PrP^Sc^ accumulation, was induced by S805. These results clearly show that in transgenic mice, as in sheep, different isolates or prion strains have different cell tropism and PrP^Sc^ processing, ultimately resulting in the differential PrP^Sc^ profiles [51,53].

Taken together, the notable contribution of the PrP^Sc^ profile in this current study has allowed us to define the ‘strain signatures’ [21] in the brains of the transgenic mouse models used here. Although the degree of prion protein expression is different, the PrP^Sc^ profile results are very close in both models, allowing discrimination of different scrapie isolates and, more importantly, discrimination of the BSE isolates. It should be noted, however, that isolates from other species may be more readily transmitted to Tgshp-IX mice, as the degree of prion protein overexpression appears to have a profound effect on the species barrier phenomenon.

In conclusion, these transgenic mouse models, as expected from their lack of species barriers due to their genetic modification, have proven efficient for prion strain typing. Furthermore, these observations corroborate the suggestion that strain typing in transgenic mouse models can be standardized using the immunohistochemical PrP^Sc^ deposition patterns [26]. However, we propose to modify the existing strain typing approach from Fraser and Dickinson [35] by (i) including additional brain sites, i.e., occipital cortex, CC, CS, granular and molecular layers of cerebellum, *fimbriae hippocampi* and (ii) the overall PrP^Sc^ reaction pattern, i.e., SBPL, PL/PL-like and ITNR/ITGL depositions.

## 4. Materials and Methods

### 4.1. TSE-Isolates

Several well-established scrapie and BSE isolates were used and had been obtained from natural and/or experimental ARQ/ARQ, ARR/ARR and VRQ/VRQ scrapie-positive sheep brain samples as well as cattle brain samples. These were:Langlade (LAN, kindly provided by Olivier Andreoletti, INRAE Toulouse, France): An upsurge of scrapie in a scrapie-unrelated sheep experimental set up in France led to the genesis of Langlade isolate in ARQ/ARQ sheep. Here, scrapie was detected in the sheep flock approximately two decades after the original initiation of the study in 1971, resulting in a comprehensive evaluation of prion disease [54].Dawson (DAW, kindly provided by M. Dawson, APHA Weybridge, Addlestone, UK): Dawson isolate had its origin from the United Kingdom in a Cheviot-Welsh sheep homozygous for the VRQ allele naturally infected with scrapie [55].S805: The S805 isolate was sourced from an ARQ/ARQ Merino sheep during follow-up surveillance of a TSE case in South Germany (Baden-Wuerttemberg). Scrapie was confirmed in this particular sheep during the active surveillance of animals for prion disease in the early 2000s throughout the European Union [56].Atypical Scrapie: The atypical scrapie isolate was confirmed during active surveillance for scrapie in an ARR/ARR sheep in the South of Germany (Baden-Wuerttemberg) in the year 2003. Unfortunately, the breed is not known.Ovine BSE (ovBSE, kindly provided by Frederic Lantier, INRAE Tours): An experimental study carried out in INRAE-Tours, France, produced the ovine BSE isolate. In the study, the investigators inoculated ARQ/ARQ New Zealand Suffolk lambs orally with sheep brain materials previously exposed intracerebrally to cattle BSE. Clinical infection was observed by the nineteenth month after inoculation [57].Classical BSE (BSE): At the FLI, Germany, an experiment investigating BSE pathogenesis in cattle generated the classical BSE isolate. Calves belonging to the Simmental cross-breed were exposed, via the oral route, to homogenates of brainstem pooled from brain samples of cattle at clinical phase of BSE infection [58]. Unfortunately, no formalin-fixed material is available from this BSE isolate in Tgshp IX mice, therefore, for lesion and PrP^Sc^ profiling an additional field Classical BSE case from the UK was used (BSE2, kindly provided by AHPA, Weybridge, UK, RQ 225:PG1199/00).

### 4.2. Mouse Experiment

The mouse experiments described here were approved (reference number 7221.3-2.1-012/03) by the competent authority of the Federal State of Mecklenburg Western Pomerania, Germany, on the basis of national and European legislation (i.e., Directive 2010/63/EU of the European Parliament and of the council on the protection of animals used for scientific purposes).

Transgenic mouse models (Tgshp IX and Tgshp XI) devoid of murine but instead overexpressing both the wild-type ARQ allele of the ovine PrP were generated at the FLI, both with the same genetic background of B6CBAx129Ola. With respect to the prion protein level of expression in the natural host species, Tgshp IX mouse line overexpresses prion protein gene in two to four folds while Tgshp XI mouse line overexpresses in four to eight folds.

Using the intracerebral route, we inoculated all mice with 25–30 μL of 10% brain tissue homogenates in 0.9% sodium chloride (as an exception atypical scrapie was used in a 1% dilution). Per isolate 10–15 mice were inoculated. Observation for development of typical clinical signs of infection was carried out at least twice a week with euthanasia of any mouse showing clinical signs suspicious of TSE, which include kyphosis, abnormal tail tonus, ataxia, paresis of the hindlimb, tremor, changes in behavior and loss of weight in several days running [59]. Besides mice, which have to be killed due to animal welfare reasons, all remaining mice were euthanized after 730 dpi at the latest.

### 4.3. Lesion Profiling

At necropsy each mouse brain was cut longitudinally and subsequently, one half was fixed in neutral buffered formalin (4%) for at least 24 h. The second half of the brain was frozen at −20 °C. Thereafter, the fixed brain was cut using the standard procedure described by [35] and processed routinely into paraffin-embedded tissue. Then, 3 µm sections were produced from the paraffin-embedded brain samples on Superfrost plus slides (Menzel, Darmstadt, Germany). Haematoxylin and eosin staining was then performed on the slide sections.

Spongiform neuropathology was evaluated according to the standard procedure described by [35] for lesion profiling in mouse model as a guide. In doing so, well-defined regions of grey and white matter from medulla oblongata, cerebellum, mesencephalon, diencephalon and telencephalon were scored. These regions are in detail

Gray matter:G1—Vestibular nuclei of medullaG2—Tectum of cerebellumG3—Cortex of the superior colliculusG4—HypothalamusG5—ThalamusG6—HippocampusG7—Septal nucleiG8—Cerebral cortex (at the level of G4 and G5)G9—Cerebral cortex (at the level of G7)

White matter:W1*—Cerebellar white matterW2*—White matter in decussation fibersW3*—Internal capsuleCC—Corpus callosum (at the level of G8)

### 4.4. PrP^Sc^-Profile

Immunodetection of PrP^Sc^ deposition in the brain samples was, with some modifications, carried out as described by [60]. Here rat anti-PrP-specific monoclonal antibody R145 (APHA Scientific) raised to the epitope 220–225 of ovine PrP protein was used as primary antibody. Briefly, slides were rehydrated through graded ethanol and then rinsed (20 min) in tap water after incubation for about 30 min in 98% formic acid. Then, 3% H_2_O_2_ in methanol was subsequently applied to inhibit endogenous peroxidase for 30 min followed by autoclaving at 121 °C in citrate buffer (pH 6.1) for 20 min. After this pretreatment process, at room temperature, we incubated slides with the R145 primary antibody diluted 1:250 in goat serum for 2 h. Prior to applying the chromogen, diaminobenzidine tetrahydrochloride (Fluka, Munich, Germany), slides were incubated in anti-rat secondary antibody (Dako, Glostrup, Denmark) for 30 min and in avidin–biotin complex (ABC, vecstatin) for 30 min hour. We then counterstained with Mayer’s hematoxylin and after dehydration coverslipped. A second section from each mouse brain was stained by diluent (goat serum) alone, serving as negative control.

Stained sections were evaluated by light microscopy. In doing so, the same regions analyzed for lesion profiling were scored with grading from weak to severe by two readers. Furthermore, abnormal reaction patterns in additional regions were documented as well.

### 4.5. Statistical Analysis

Statistical analysis comparing the plaque/plaque-like formation or subpial reaction pattern, respectively, of the different prion strains in the tissues studied, was performed using a two-way ANOVA test for multiple comparisons with Sidák correction (GraphPad Prism 9). The analysis was performed with α = 0.05; *p* values are graphically represented in Appendix A by asterisk (*p* < 0.01: **; *p* < 0.001: ***; *p* < 0.0001: ****).

## Figures and Tables

**Figure 1 ijms-23-06744-f001:**
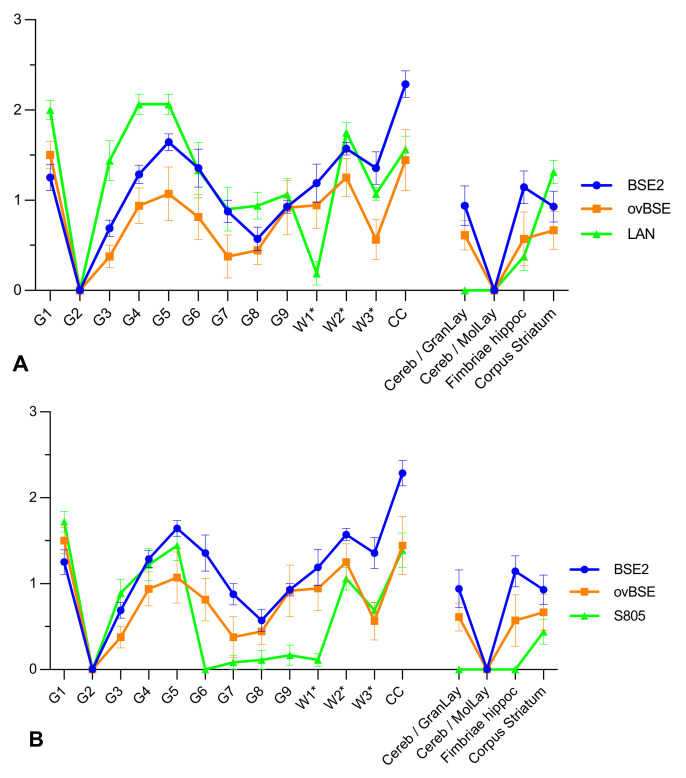
Tgshp IX PrP^Sc^ profiles of the different scrapie isolates compared to BSE2 and ovBSE. (**A**) LAN (**B**) S805, (**C**) DAW, (**D**) Atypical scrapie versus BSE2 and ovBSE; Note that BSE and ovBSE revealed a similar pattern. All scrapie isolates can be discriminated in the Tgshp IX mouse model. Grey matter region, G1 = vestibular nuclei; G2 = tectum; G3 = cortex of the superior colliculus; G4 = hypothalamus; G5 = thalamus; G6 = hippocampus; G7 = septal nuclei; G8 = cerebral cortex (at the level of G4 and G5); G9 = cerebral cortex (at the level of G7); for white matter, W1* = cerebellar white matter; W2* = white matter in decussation fibers; W3* = internal capsule; CC = corpus callosum (at the level of G8); additional regions Cereb/GranLay = Granular layer of cerebellum; Cereb/MolLay = Molecular layer of cerebellum; Fimbriae hippoc. = Fimbriae hippocampi. Error bars = standard error of means.

**Figure 2 ijms-23-06744-f002:**
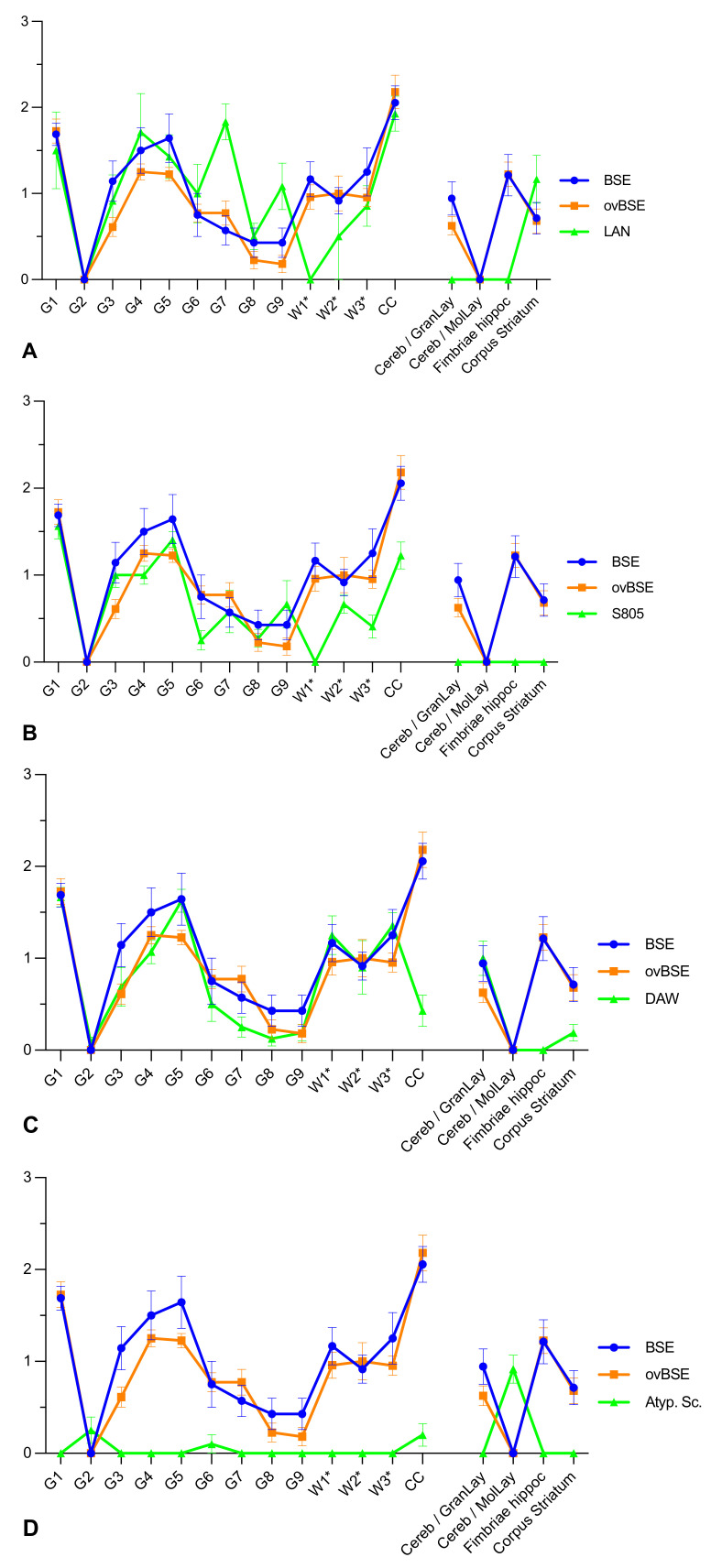
Tgshp XI PrP^Sc^ profiles of the different scrapie isolates compared to BSE and ovBSE. (**A**) LAN (**B**) S805, (**C**) DAW, (**D**) Atypical scrapie versus BSE and ovBSE; Note that BSE and ovBSE revealed a similar pattern. All scrapie isolates can be discriminated in the Tgshp XI mouse model. Grey matter region, G1 = vestibular nuclei; G2 = tectum; G3 = cortex of the superior colliculus; G4 = hypothalamus; G5 = thalamus; G6 = hippocampus; G7 = septal nuclei; G8 = cerebral cortex (at the level of G4 and G5); G9 = cerebral cortex (at the level of G7); for white matter, W1* = cerebellar white matter; W2* = white matter in decussation fibers; W3* = internal capsule; CC = corpus callosum (at the level of G8); additional regions Cereb/GranLay = Granular layer of cerebellum; Cereb/MolLay = Molecular layer of cerebellum; Fimbriae hippoc. = Fimbriae hippocampi. Error bars = standard error of means.

**Figure 3 ijms-23-06744-f003:**
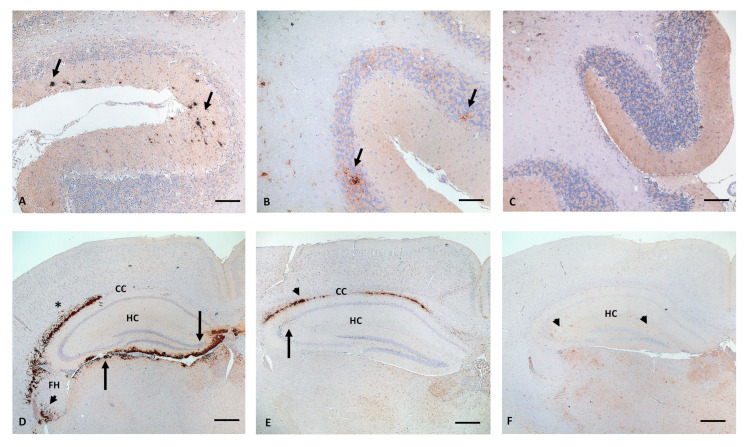
Characteristic PrP^Sc^-profile in cerebellum (**A**–**C**) and Hippocampus (**D**,**E**) of Tgshp IX mice inoculated with different isolates. (**A**) atypical scrapie: multifocal distinct PrP^Sc^ accumulations (arrows) solely in the molecular layer are unique; (**B**) ovBSE: accumulation of PrP^Sc^ in granular layer (arrows) and white matter of cerebellum; (**C**) S805: only minor amounts of PrP^Sc^ are seen in white matter but no accumulation can be found in molecular or granular layer (similar pattern is seen with LAN); (**D**) ovBSE: severe plaques, plaque-like and coalescing granular PrP^Sc^ accumulation in corpus callosum (CC, *) and fimbriae hippocampi (FH, arrowhead), additionally the subpial reaction pattern in thalamus and hippocampus (arrows) is a typical characteristic of the BSE isolates; (**E**) LAN: CC with moderate coalescing PrP^Sc^ deposition (arrowhead), as well as focal granular reaction pattern in hippocampus (arrow); (**F**) DAW: no PrP^Sc^ in CC, multifocal mild granular PrP^Sc^ deposition (arrowheads) in hippocampus (HC) instead; Immunohistochemistry, monoclonal anti-PrP R145, Bar 100 µm.

**Figure 4 ijms-23-06744-f004:**
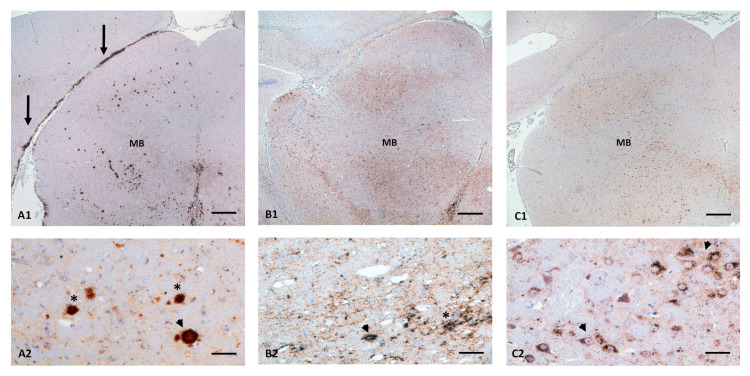
Characteristic PrP^Sc^-profile in midbrain (**A1**–**C1**) as well as distinct isolate dependent cellular reaction pattern (**A2**–**C2**). (**A1**,**A2**) ovBSE: predominant plaque (arrowhead) and plaque-like (*) PrP^Sc^ deposition pattern in midbrain (MB), additionally massive subpial PrP^Sc^ accumulation are visible in occipital cortex (arrows), both reaction pattern are typical for BSE isolates; (**B1**,**B2**) LAN: a variable reaction pattern is seen in MB, with a predominant diffuse coarse to coalescing granular reaction pattern (*) and even a multifocal perivascular staining reaction (arrowhead); (**C1**,**C2**) S805: a very prominent intraneuronal (arrowhead) PrP^Sc^ accumulation is characteristic for this isolate; Immunohistochemistry, monoclonal anti-PrP R145, Bar (**A1**–**C1**) 100 µm (**A2**–**C2**) 20 µm.

**Table 1 ijms-23-06744-t001:** Comparison of the most important results obtained by passaging different BSE and scrapie isolates in Tgshp IX and Tgshp XI mice.

		Tgshp IX	Tgshp XI
**BSE ***	Incubation time	351 ± 165 dpi (BSE) and 262 ± 21 dpi (BSE2)	445 ± 185 dpi
	Attack rate	6/6 (BSE2 8/8)	9/9
	Lesion Profile	PL in CC	PL in CC
	PrP^Sc^ profile	1, 4, 5, 6, 2*	1, 4, 5, 1*, 2*, 3*
	Unique pattern	Severe in CC (with PL/PL-like)	Severe in CC (with PL/PL-like)
		Granular Layer of Cerebellum	Granular Layer of Cerebellum
		Strong in Fimbriae hippocampi	Strong in Fimbriae hippocampi
		Severe/widespread SBPL reaction pattern	Severe/widespread SBPL reaction pattern
		Widespread PL/PL-like deposition	Widespread PL/PL-like deposition
		Multifocal deposition in Thalamus	Multifocal deposition in Thalamus
**ovBSE**	Incubation time	230 ± 63 dpi	231 ± 9 dpi
	Attack rate	9/9	12/12
	Lesion Profile	PL in CC	PL in CC
	PrP^Sc^ profile	1, 4, 5, 2*	1, 4, 5, 1*, 2*, 3*
	Additional Characteristics	Severe in CC (with PL/PL-like)	Severe in CC (with PL/PL-like)
		Cerebellar White Matter and Granular Layer	Cerebellar White Matter and Granular Layer
		Strong in Fimbriae hippocampi	Strong in Fimbriae hippocampi
		Severe/widespread SBPL reaction pattern	Severe/widespread SBPL reaction pattern
		Widespread PL/PL-like deposition	Widespread PL/PL-like deposition
		Multifocal deposition in Thalamus	Multifocal deposition in Thalamus
**LAN**	Incubation time	190 ± 3 dpi	195 ± 11 dpi
	Attack rate	8/8	8/9
	Lesion Profile	PL in CC	PL in CC
	PrP^Sc^ profile	1, 4, 5, 2* none in 1*	1, 4, 7, 9, 3*, none in 1*
	Additional Characteristics	Severe in CC (coalescing)	Severe in CC (PL, Pl-like)
		Distinct in Corpus striatum	Distinct in Corpus striatum
		Mild/oligofocal SBPL reaction pattern	Mild oligofocal SBPL reaction pattern
		Mild PL-like	Mild PL-like, PL
		Multifocal deposition in Thalamus	Multifocal deposition in Thalamus
**S805**	Incubation time	160 ± 18 dpi	158 ± 20 dpi
	Attack rate	9/9	12/12
	Lesion Profile	PL in CC	PL in CC
	PrP^Sc^ profile	1, 4, 5, 2*, none in 6–9, 1*	1, 3, 4, 5, 2*, none in 1*
	Additional Characteristics	Severe in CC (with PL/PL-like)	Severe in CC (with PL/PL-like)
		Hippocampus devoid of PrP^Sc^	Hippocampus weakly affected (single mice)
		SBPL reaction pattern rarely seen	SBPL reaction pattern rarely seen
		Prominent ITNR/ITGL	Prominent ITNR/ITGL
		Multifocal deposition in Thalamus	Only dorsolateral nuclei of Thalamus involved
**DAW**	Incubation time	371 ± 44 dpi	342 ± 39 dpi
	Attack rate	9/10	8/9
	Lesion Profile	No PL	No PL
	PrP^Sc^ profile	1, 4, 5, 1*, 2*, 3*, very weak in 7–9	1, 5, 1*, 2*, 3*
	Additional Characteristics	Weak (if any) accumulation in CC	Weak (if any) accumulation in CC
		Cerebellar White Matter	Cerebellar White Matter and Granular Layer
		SBPL reaction pattern rarely seen	SBPL reaction pattern rarely seen
		Devoid of PL/PL-like formation	Devoid of PL/PL-like formation
		Diffuse deposition in Thalamus	Multifocal deposition in Thalamus
**Atyp. Sc.**	Incubation time	317 ± 4,5 dpi	272 ± 32 dpi
	Attack rate	2/2	6/6
	Lesion Profile	No PL	No PL
	PrP^Sc^ profile	1*, none in all other regions	2, 6
	Additional Characteristics	Cerebellar Molecular Layer	Cerebellar Molecular Layer
		None in CC	Mild in CC
		Hippocampus devoid of PrPSc	Hippocampus devoid of PrPSc
		SBPL reaction pattern rarely seen	SBPL reaction pattern rarely seen
		Mild PL-like in Cerebellum	Mild PL-like in Cerebellum
		None in Thalamus	None in Thalamus

Legend: dpi = day post infection; 1 = Nucleus vestibularis; 2 = Tectum; 3 = Colliculus superficialis dorsalis; 4 = Hypothalamus; 5 = Thalamus; 6 = Hippocampus; 7 = Septum; 9 = Cortex B; 1* = Cerebllar White Matter; 2* = Decussatio; 3* = Capsula Interna; CC = Corpus Callosum; SBPL = Subpial; PL = Plaque; ITNR = Intraneuronal; ITGL = Intraglial; * two BSE isolates were used, BSE was inoculated into both mouse models, but no formalin fixed material was available for comparison, therefore BSE2 inoculated in Tgshp IX mice was additionally used.

## Data Availability

Not applicable.

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
