# Peer review of "Strain Typing of Classical Scrapie and Bovine Spongiform Encephalopathy (BSE) by Using Ovine PrP (ARQ/ARQ) Overexpressing Transgenic Mice"

_ijms, 2022, doi:10.3390/ijms23126744_

Round 1
Reviewer 1 Report
Using transgenic mice overexpressing ovine PrPC is promising idea for the future development of strain tying of animal prions. Since at this moment we have many kinds of animal prions especially wild animals. I hope author could extend their projects many different fields.
There is important argue about species barrier phenomenon during prion transmission. If we could reduce this donor-recipient effect during transmission, then we could evaluate the transmission effect more clearly. In this study ovine prion transgenic mice are quite promising, not just for characterizing scrapie agent but also for BSE agent. However, I found out some questions in this manuscript.
Major questions.
L183; 'by DAW' could be 'by DAW and S805 (Figure 3C)'.
L188; 'cerebellum (C) S805' could be 'cerebellum; (C) S805'.
Minor point.
L11; 'pathological' could be 'abnormal'.
L33; 'man' could be 'human'.
L37; 'non-contagious' could be 'sporadic and non-contagious'.
L52; 'scrapie' could be 'scrapie agents'.
L61; 'of different' could be 'from different'.
L111; '272+32 dpi' could be '272+32 dpi with Atyp. Sc.' .
L163; 'in CC' could be 'in CC with atypical scrapie isolates'.
This is very important paper. Without mistype we could understand these easier.
I think this manuscript is eligible to publish from MDPI journal.
Reviewer 2 Report
This article describes in details the interest ovinized transgenic mouse models as bioassay to type different prion strains. The study is made by distinguished scientists. The presentation is clear.
I have only one major comment. The strain typing shown here is solely based on comparing the disease tempo of inoculated animals and the neuropathological features. Why did the authors exclude western blot analyses to type PrPSc electrophoretic signature? This would allow comparing the signature in natural host versus ovinized mice (e.g. is atypical scrapie still atypical in these mice?). Overall this would greatly strengthen their analyses.
Round 2
Reviewer 2 Report
The authors have addressed my concern.